# Prediction of preterm birth in nulliparous women using logistic regression and machine learning

**Reza Arabi Belaghi[1,2], Joseph Beyene[3,4], Sarah D. McDonald[1,3,5,6]** *

**1** Department of Obstetrics and Gynecology, McMaster University, Hamilton, Ontario, Canada,
**2** Department of Statistics, University of Tabriz, Tabriz, Iran, **3** Department of Health Research Methods, Evidence & Impact, McMaster University, Hamilton, Ontario, Canada, **4** Department of Mathematics and Statistics, McMaster University, Hamilton, Ontario, Canada, **5** Department of Obstetrics and Gynecology (Division of Maternal-Fetal Medicine), McMaster University, Hamilton, Ontario, Canada, **6** Department of Radiology, McMaster University, Hamilton, Ontario, Canada

* mcdonals@mcmaster.ca

## Abstract

### Objective

To predict preterm birth in nulliparous women using logistic regression and machine learning.

### Design

Population-based retrospective cohort.

### Participants

Nulliparous women (N = 112,963) with a singleton gestation who gave birth between 20–42 weeks gestation in Ontario hospitals from April 1, 2012 to March 31, 2014.

### Methods

We used data during the first and second trimesters to build logistic regression and machine learning models in a "training" sample to predict overall and spontaneous preterm birth. We assessed model performance using various measures of accuracy including sensitivity, specificity, positive predictive value, negative predictive value, and area under the receiver operating characteristic curve (AUC) in an independent "validation" sample.

### Results

During the first trimester, logistic regression identified 13 variables associated with preterm birth, of which the strongest predictors were diabetes (Type I: adjusted odds ratio (AOR): 4.21; 95% confidence interval (CI): 3.23–5.42; Type II: AOR: 2.68; 95% CI: 2.05–3.46) and abnormal pregnancy-associated plasma protein A concentration (AOR: 2.04; 95% CI: 1.80–2.30). During the first trimester, the maximum AUC was 60% (95% CI: 58–62%) with artificial neural networks in the validation sample. During the second trimester, 17 variables

**Data Availability Statement:** The data underlying this study are not publicly available due to a legally-binding Data Use Agreement that restricts our ability to share the data. Therefore, as per the signed agreement with the BORN database, only

Authorized Users are permitted access to the data and a Signed Confidentiality Agreement is required. BORN Ontario is a prescribed registry established in Ontario under the Personal Health Information Protection Act, 2004 (PHIPA) for the purpose of facilitating and/or improving the provision of health care in Ontario, with a vision for the best possible beginnings for lifelong health. Policies regarding data access can be found at https://www.bornontario.ca/en/data/requesting-data.aspx. Please contact BORN for further information here: Science@BORNOntario.ca. For information regarding Data Privacy and Security please contact BORN Ontario Privacy Officer, directed here: Privacy@BORNOntario.ca.

**Funding:** This work was supported by the Canadian Institutes of Health Research (CIHR; grant #: 151520). Dr. McDonald is supported by a Tier II CIHR Canada Research Chair (950-229920). Joseph Beyene holds the John D. Cameron Endowed Chair in the Genetic Determinants of Chronic Diseases, McMaster University. CIHR had no role in the design or conduct of the study; collection, management, analysis, and interpretation of the data; preparation, review, or approval of the manuscript; or the decision to submit the manuscript for publication.

**Competing interests:** The authors have no conflicts of interest to declare.

were significantly associated with preterm birth, among which complications during pregnancy had the highest AOR (13.03; 95% CI: 12.21–13.90). During the second trimester, the AUC increased to 65% (95% CI: 63–66%) with artificial neural networks in the validation sample. Including complications during the pregnancy yielded an AUC of 80% (95% CI: 79–81%) with artificial neural networks. All models yielded 94–97% negative predictive values for spontaneous PTB during the first and second trimesters.

## Conclusion

Although artificial neural networks provided slightly higher AUC than logistic regression, prediction of preterm birth in the first trimester remained elusive. However, including data from the second trimester improved prediction to a moderate level by both logistic regression and machine learning approaches.

## Introduction

Preterm birth (PTB), birth before 37 weeks, is the leading cause of neonatal death and disability [1]. Approximately, 50% of all perinatal deaths are caused by PTB [2]. In the U.S., almost 10% of babies are born preterm [3], costing the healthcare system at least $26 billion yearly [4]. In Canada, PTB comprises 8% of all births and results in direct costs of $580 million annually [5]. Risk factors for PTB are heterogeneous and include previous PTB, race, age, nulliparity, urinary tract infection, smoking, and bleeding during early pregnancy [6–8]. Prediction of PTB would facilitate the use of therapeutic interventions to reduce infant morbidity and mortality, thereby benefitting families, society, and the healthcare system.

Previous studies have found the prediction of PTB to be challenging, whether by logistic regression or machine learning. The area under the receiver operating characteristic curve (AUC) for prediction of PTB in previous studies ranged from 62% to 72% depending on the number of predictors and study design [9–15]. The predictive power of the machine learning model developed by Fergus *et al.* [16] was promising (AUC, 95%), but measuring uterine electrical signals (electrohysterography) is not practical on a large scale. Another drawback was the synthetic oversampling of the whole dataset, rather than just the training dataset, thereby calling into question the 95% AUC of that work.

Machine learning is a computer programming approach whereby computers learn from "big data" to make better predictions [17]. In 2019, machine learning was identified as one of the most advanced tools for prenatal diagnosis [18]. Morover, machine learning has been broadly applied in medicine, from cancer detection [19, 20] to prediction of cardiovascular diseases [21], among others. In this study, we considered some of state-of-the-art machine learning methods, including decision trees, random forests, and artificial neural networks, that are frequently used in medicine to develop prediction models [21–28]. We also considered logistic regression as a traditional statistical approach to develop prediction models [29]. Unlike logistic regression, machine learning approaches are free of statistical assumptions (such as linearity and uncorrelated predictors) and can handle complex interactions between predictive factors without these interactions being explicitly specified [27, 30].

We aimed to overcome the challenges of predicting PTB, especially for nulliparous women, by evaluating logistic regression and multiple machine learning algorithms. To this end, we considered variables available in clinical care, including some not previously assessed in other

studies. Our study aimed to: 1) identify important predictors associated with PTB during the first and second trimester in nulliparous women from a large population cohort; and 2) construct models to predict PTB based on logistic regression and robust machine learning algorithms.

## Methods and materials

### Data and population

Ontario comprises 40% of the Canadian population and has approximately 140,000 births each year [31]. We performed a population-based retrospective cohort study using Ontario's Better Outcomes Registry and Network (BORN) database, which includes a wide range of maternal, antenatal, and birth data [32]. We included all nulliparous women with singleton births who gave birth between 20 and 42 weeks gestation in an Ontario hospital between April 1, 2012 and March 31, 2014.

**Outcome.** PTB was the primary outcome variable in this study, defined as gestational age at birth (from ultrasound estimation or calculation from the first day of the last menstrual period) <37 weeks. We also considered spontaneous PTB as a secondary outcome. Spontaneous PTB was identified using the definition of Maghsouldu *et al.* [33], i.e.: not "induced", not "caesarean section" and not "augmented labor".

**Predictors.** We considered predictors based on our literature review of PTB risk factors during the first and second trimesters [7, 34]. We considered socio-demographic variables including maternal age, height, pre-pregnancy body mass index (BMI), gestational weight gain during the first trimester, income, education, race, and immigration status. Further, we included the number of previous abortions (which includes miscarriages), conception type, smoking status, alcohol consumption, folic acid use, pre-existing medical health conditions, diabetes, pre-existing mental health conditions (such as anxiety, depression, and addiction) and antenatal health care provider type.

Pregnancy-associated plasma protein A and free beta-subunit of human chorionic gonadotropin were measured during the first trimester as part of the screen for Down syndrome [30], but we considered them as potential markers of placental and preeclamptic diseases [35]. We also included ultrasound measurement of nuchal translucency as another predictor [36]. For the second-trimester models, we included all of the predictors from the first trimester plus information that became available during the second trimester including dimeric inhibin A, unconjugated estriol, human chorionic gonadotropin, alpha-fetoprotein concentration, hypertensive disorders of pregnancy, gestational diabetes, infections, medication exposure, sex of the fetus, and complications during pregnancy [37].

We grouped maternal height into four categories, including <150 cm, 150 cm—169 cm, 160 cm—169 cm, and ≥170 cm. We classified pre-pregnancy BMI as underweight (<18.5 kg/m$^2$), normal weight (18.5–24.9 kg/m$^2$), overweight (25–29.9 kg/m$^2$), and obese (≥30 kg/m$^2$), according to World Health Organization criteria [38, 39]. We used the Institute of Medicine guidelines [40] to categorize gestational weight gain into three groups, including recommended weight gain, less than recommended weight gain, and more than recommended weight gain. For income, education, race, and immigration status, we used neighbourhood income quartiles, neighbourhood education quartiles, neighbourhood immigrant concentration, and neighbourhood minority quartiles, respectively (see S1 Table for the definition of these variables).

We categorized the number of previous abortions (including spontaneous and therapeutic abortions) into four groups based on Oliver *et al.* [41], including 0, 1, 2, and 3+. We grouped the pre-existing health conditions variable in the BORN database into "Yes" or "No" since that

variable had more than 1000 possible entries (S2 Table). We treated pre-existing mental health conditions (S3 Table) as a binary categorical variable. We classified the conception type into: spontaneous, *in vitro* fertilization (IVF, or a combination of IVF and other methods), and other methods (such as Surrogate, Intrauterine insemination alone, or unknown) [42].

We classified protein concentrations (pregnancy-associated plasma protein A, free beta-subunit of human chorionic gonadotropin, dimeric inhibin A, unconjugated estriol, human chorionic gonadotropin, and alpha-fetoprotein) and nuchal translucency as normal, abnormal, and missing (cut-off values shown in S4 Table). The variable "complications during pregnancy" had more than 600 categories, and we therefore categorized data for this variable into three groups based on maternal-fetal expertise (SDM) as follows: no complications, mild-moderate complications, and severe complications [37].

### Statistical analysis

We used the Chi-square test and univariate logistic regression to measure associations between predictors and PTB. We assessed statistical significance using 2-sided p-values, with a p-value <0.05 considered statistically significant. We then proceed with variable selection using stepwise multivariable logistic regression based on the Akaike Information Criterion (AIC). We also utilized the Boruta algorithm to select important variables for the machine learning models [43]. In short, Boruta is based on the random forest machine learning method, which selects relevant variables that significantly impact the prediction power of the model [43].

We followed the guidelines for the Transparent Reporting of a Multivariable Prediction Model for Individual Prognosis or Diagnosis [44] for establishing prediction models. Based on these guidelines, we selected 2/3 of the data as the training set and the remaining 1/3 of the data as the test (validation) set. We balanced the training samples using a random over-sampling technique [45]. We then used ten-fold cross-validation to establish machine learning models. Finally, we used the test data to evaluate the performance of the proposed prediction models by comparing the sensitivity, specificity, positive predictive values, negative predictive values, and AUC. We performed all machine learning computations in *R* software using the *caret* package [46].

We applied multiple imputation with 10 imputations [47–49] to replace missing observations on the predictors. However, for plasma proteins and nuchal translucency, missing data were treated as a new category since a large proportion of women chose not to enroll in screening for Down syndrome. We also treated gestational weight gain during the first trimester in a similar manner, since the lack of recording of weight gain may reflect less than optimal care. The Hamilton Integrated Research Ethics Board approved the study before study commencement (approval #: 14-714-C).

## Results

### Study participants and univariate analysis

Of 112,963 nulliparous women with singleton pregnancies, PTB occurred in 6,955 (6.2%, Table 1). Out of all PTBs, there were 3,695 (53%) spontaneous PTBs. Approximately 5% of patients were younger than 20 years of age, while 13% were over age 35 years. Approximately 2% of patients had three or more previous abortions including miscarriages. More than 50% of patients had a non-ideal pre-pregnancy BMI, of which 17.34% and 12.58% were overweight and obese, respectively. Approximately 17% of the cohort had at least one pre-existing medical condition. Only 78.67% of the patients had a documented first-trimester appointment.

During the first trimester, we examined 23 predictors (Table 2). Women who were under 25 years of age, shorter in stature (<160 cm), had pre-pregnancy obesity, conceived with IVF,

**Table 1. Distribution of maternal baseline characteristics, demographics, and clinical variables in nulliparous women.**

| Variables | Levels | N | % |
|---|---|---|---|
| Age (years) | <20 | 5782 | 5.12 |
| | 20–24 | 17979 | 15.92 |
| | 25–29 | 36309 | 32.14 |
| | 30–34 | 34798 | 30.80 |
| | 35+ | 14817 | 13.12 |
| | Missing | 3278 | 2.90 |
| Height | <150 cm | 2663 | 2.36 |
| | 150 cm-159 cm | 21714 | 19.22 |
| | 160 cm-169 cm | 51090 | 45.23 |
| | ≥170 cm | 22662 | 20.06 |
| | Missing | 14834 | 13.13 |
| | Mean = 163.7, SD = 7.34 | | |
| Pre-pregnancy body mass index (kg/m$^2$) | Normal | 51225 | 45.35 |
| | Overweight | 19584 | 17.34 |
| | Obese | 14212 | 12.58 |
| | Underweight | 5929 | 5.25 |
| | Missing | 22013 | 19.49 |
| | Mean = 24.9, SD = 6.29 | | |
| Neighbourhood income quartile | First quartile (lowest) | 29891 | 26.46 |
| | Second quartile | 25117 | 22.23 |
| | Third quartile | 26122 | 23.12 |
| | Fourth quartile (highest) | 27466 | 24.31 |
| | Missing | 4367 | 3.87 |
| Neighbourhood education quartile | First quartile (lowest) | 27849 | 24.65 |
| | Second quartile | 28552 | 25.28 |
| | Third quartile | 28089 | 24.87 |
| | Fourth quartile (highest) | 24980 | 22.11 |
| | Missing | 3493 | 3.09 |
| Neighbourhood minority quartile | First quartile (lowest) | 23762 | 21.04 |
| | Second quartile | 18718 | 16.57 |
| | Third quartile | 23705 | 20.98 |
| | Fourth quartile (highest) | 43285 | 38.32 |
| | Missing | 3493 | 3.09 |
| Neighbourhood immigration quartile | First quartile (lowest) | 24129 | 21.36 |
| | Second quartile | 20274 | 17.95 |
| | Third quartile | 24785 | 21.94 |
| | Fourth quartile (highest) | 39937 | 35.35 |
| | Missing | 3838 | 3.40 |
| Smoking status | Non-smoker | 97265 | 86.10 |
| | Smoker | 10986 | 9.73 |
| | Missing | 4712 | 4.17 |
| Ex-smoker | No | 71466 | 63.26 |
| | Yes | 16153 | 14.30 |
| | Missing | 25344 | 22.44 |
| Alcohol consumption | No | 101902 | 90.21 |
| | Yes | 2185 | 1.93 |
| | Missing | 8876 | 7.86 |

*(Continued)*

**Table 1.** (Continued)

| Variables | Levels | N | % |
|---|---|---|---|
| Drug (substance) use | No | 102688 | 90.90 |
| | Yes | 2555 | 2.26 |
| | Missing | 7720 | 6.83 |
| First-trimester visit | Yes | 88866 | 78.67 |
| | No | 10983 | 9.72 |
| | Unknown | 13114 | 11.61 |
| Antenatal health care provider | Obstetrician | 98471 | 87.17 |
| | Midwife | 13561 | 12.00 |
| | Missing | 931 | 0.82 |
| Folic acid use | Yes | 78617 | 69.60 |
| | No | 21199 | 18.77 |
| | Missing | 13147 | 11.64 |
| Intention to breastfeed | Yes | 101057 | 89.46 |
| | No | 4933 | 4.37 |
| | Missing | 6973 | 6.17 |
| Pre-existing health conditions | No | 88390 | 78.25 |
| | Yes | 19608 | 17.36 |
| | Missing | 4965 | 4.40 |
| Pre-existing mental health conditions | No | 91666 | 81.15 |
| | Yes | 14932 | 13.22 |
| | Missing | 7720 | 6.83 |
| Number of previous abortions (including miscarriages) | 0 | 80615 | 71.36 |
| | 1 | 19189 | 16.99 |
| | 2 | 5334 | 4.72 |
| | 3+ | 2299 | 2.04 |
| | Missing | 5526 | 4.89 |
| Conception type | Spontaneous | 105061 | 93.00 |
| | IVF and combination | 2176 | 1.93 |
| | Other | 2662 | 2.36 |
| | Missing | 3064 | 2.71 |
| Gravidity | Mean = 1.38, SD = 0.84 | | |
| Diabetes | No diabetes | 102308 | 90.57 |
| | Type I | 356 | 0.32 |
| | Type II | 454 | 0.40 |
| | Missing | 9845 | 8.72 |
| Gestational weight gain during the first trimester | Recommended | 10034 | 8.88 |
| | <Recommended | 20477 | 18.13 |
| | >Recommended | 18842 | 16.68 |
| | Missing | 63610 | 56.31 |
| Pregnancy-associated plasma protein A | Normal | 60121 | 53.22 |
| | Abnormal | 3126 | 2.77 |
| | Missing | 49716 | 44.01 |
| Free beta-subunit of human chorionic gonadotropin | Normal | 105928 | 93.77 |
| | Abnormal | 6350 | 5.62 |
| | Missing | 685 | 0.61 |

(*Continued*)

**Table 1.** (Continued)

| Variables | Levels | N | % |
|---|---|---|---|
| Nuchal translucency | Normal | 50550 | 44.75 |
| | Abnormal | 47 | 0.04 |
| | Missing | 62366 | 55.21 |
| Dimeric inhibin A | Normal | 7746 | 6.86 |
| | Abnormal | 564 | 0.50 |
| | Missing | 104653 | 92.64 |
| Unconjugated estriol | Normal | 61445 | 54.39 |
| | Abnormal | 290 | 0.26 |
| | Missing | 51228 | 45.35 |
| Human chorionic gonadotropin | Normal | 60733 | 53.76 |
| | Abnormal | 899 | 0.80 |
| | Missing | 51331 | 45.44 |
| Alpha-fetoprotein | Normal | 60610 | 53.65 |
| | Abnormal | 1616 | 1.44 |
| | Missing | 50737 | 44.9 |
| Diabetes during the second trimester | No diabetes | 97048 | 85.91 |
| | Gestational diabetes | 5228 | 4.63 |
| | Type I | 356 | 0.32 |
| | Type II | 454 | 0.40 |
| | Type unknown | 32 | 0.03 |
| | Missing | 9845 | 8.72 |
| Hypertensive disorder | None | 99619 | 88.19 |
| | Eclampsia | 63 | 0.06 |
| | Gestational hypertension | 5267 | 4.66 |
| | HELLP | 179 | 0.16 |
| | Preeclampsia | 914 | 0.81 |
| | Unknown | 6921 | 6.13 |
| Infection(s) | No | 80156 | 70.96 |
| | Yes | 24697 | 21.86 |
| | Missing | 8110 | 7.18 |
| Medication exposure | No | 20743 | 18.36 |
| | Vitamin and herbals | 50410 | 44.63 |
| | Other medication | 30384 | 26.90 |
| | Missing | 11426 | 10.11 |
| Sex of fetus | Female | 54612 | 48.35 |
| | Male | 58065 | 51.40 |
| | Missing | 286 | 0.25 |
| Complications during pregnancy | No complications | 90302 | 79.94 |
| | Mild-moderate complications | 4676 | 4.14 |
| | Severe complications | 14255 | 12.62 |
| | Missing | 3730 | 3.30 |

Preterm birth: n = 6,955 (6.16%); Spontaneous PTB: n = 3695 (5.62%); Term birth: n = 106,008 (93.84%); SD: Standard deviation; IVF: *In vitro* fertilization; Pre-existing maternal health conditions shown in S2 Table. Pre-existing mental health conditions shown in S3 Table.

had prior medical conditions including diabetes, and those with low pregnancy-associated plasma protein A concentrations were more likely than women without these conditions to

**Table 2. Univariate analyses of associations between each predictor and preterm birth during the first trimester in nulliparous women.**

| Variables | Levels | Term birth 85457 (93.8%) | | Preterm birth 5645 (6.2%) | | Chi-square test | | |
|---|---|---|---|---|---|---|---|---|
| | | N | % | N | % | P-Value | Crude OR | 95% CI |
| Age (years) | <20 | 4149 | 4.86 | 223 | 3.95 | <0.001 | 1.20 | (1.05–1.39) |
| | 20–24 | 13874 | 16.24 | 791 | 14.01 | | 1.24 | (1.04–1.24) |
| | 25–29 | 29361 | 34.36 | 1908 | 33.80 | | Reference | |
| | 30–34 | 27310 | 31.96 | 1897 | 33.60 | | 0.93 | (0.87–0.99) |
| | 35+ | 10763 | 12.59 | 826 | 14.63 | | 0.84 | (0.77–0.92) |
| Height | <150 cm | 1963 | 2.30 | 172 | 3.05 | <0.001 | 1.33 | (1.13–1.55) |
| | 150 cm-159 cm | 17588 | 20.58 | 1395 | 24.71 | | 1.20 | (1.12–1.29) |
| | 160 cm-169 cm | 46763 | 54.72 | 3085 | 54.65 | | Reference | |
| | ≥170 cm | 19143 | 22.40 | 993 | 17.59 | | 0.79 | (0.73–0.84) |
| Pre-pregnancy body mass index (kg/m²) | Normal | 52107 | 60.97 | 3245 | 57.48 | <0.001 | Reference | |
| | Overweight | 16434 | 19.23 | 1103 | 19.54 | | 1.07 | (1.00–1.15) |
| | Obese | 12315 | 14.41 | 983 | 17.41 | | 1.28 | (1.18–1.38) |
| | Underweight | 4601 | 5.38 | 314 | 5.56 | | 1.09 | (0.97–1.23) |
| Neighbourhood income quartile | First quartile (lowest) | 22363 | 26.17 | 1481 | 26.24 | 0.87 | 0.98 | (0.91–1.06) |
| | Second quartile | 19930 | 23.32 | 1341 | 23.76 | | Reference | |
| | Third quartile | 21431 | 25.08 | 1401 | 24.82 | | 0.97 | (0.90–1.05) |
| | Fourth quartile (highest) | 21733 | 25.43 | 1422 | 25.19 | | 0.97 | (0.90–1.04) |
| Neighbourhood education quartile | First quartile (lowest) | 20734 | 24.26 | 1302 | 23.06 | 0.029 | 0.98 | (0.90–1.05) |
| | Second quartile | 23152 | 27.09 | 1490 | 26.40 | | Reference | |
| | Third quartile | 22149 | 25.92 | 1493 | 26.45 | | 1.04 | (0.97–1.12) |
| | Fourth quartile (highest) | 19422 | 22.73 | 1360 | 24.09 | | 1.08 | (1.01–1.17) |
| Neighbourhood minority quartile | First quartile (lowest) | 20505 | 23.99 | 1415 | 25.07 | 0.048 | 1.01 | (0.93–1.09) |
| | Second quartile | 15694 | 18.36 | 1071 | 18.97 | | Reference | |
| | Third quartile | 17916 | 20.96 | 1186 | 21.01 | | 0.97 | (0.89–1.05) |
| | Fourth quartile (highest) | 31342 | 36.68 | 1973 | 34.95 | | 0.92 | (0.85–0.99) |
| Neighbourhood immigration quartile | First quartile (lowest) | 21124 | 24.72 | 1518 | 26.89 | 0.001 | 1.11 | (1.02–1.20) |
| | Second quartile | 16978 | 19.87 | 1098 | 19.45 | | Reference | |
| | Third quartile | 18742 | 21.93 | 1253 | 22.20 | | 1.03 | (0.95–1.12) |
| | Fourth quartile (highest) | 28613 | 33.48 | 1776 | 31.46 | | 0.95 | (0.88–1.03) |
| Ex-smoker | No | 70981 | 83.06 | 4632 | 82.05 | 0.054 | Reference | |
| | Yes | 14476 | 16.94 | 1013 | 17.95 | | 1.07 | (0.99–1.14) |
| Smoking status | Non-smoker | 76892 | 89.98 | 5017 | 88.88 | 0.008 | Reference | |
| | Smoker | 8565 | 10.02 | 628 | 11.12 | | 1.12 | (1.03–1.22) |
| Folic acid use | Yes | 68486 | 80.14 | 4610 | 81.67 | 0.006 | Reference | |
| | No | 16971 | 19.86 | 1035 | 18.33 | | 0.90 | (0.84–0.97) |
| Conception type | Spontaneous | 81713 | 95.62 | 5276 | 93.46 | <0.001 | Reference | |
| | *In vitro* fertilization and combination | 1536 | 1.80 | 204 | 3.61 | | 2.07 | (1.76–2.38) |
| | Other | 2208 | 2.58 | 165 | 2.92 | | 1.15 | (0.98–1.35) |
| Number of previous abortions | 0 | 64133 | 75.05 | 4113 | 72.86 | <0.001 | Reference | |
| | 1 | 15254 | 17.85 | 1048 | 18.57 | | 1.07 | (0.99–1.14) |
| | 2 | 4268 | 4.99 | 313 | 5.54 | | 1.14 | (1.01–1.28) |
| | 3+ | 1802 | 2.11 | 171 | 3.03 | | 1.48 | (1.25–1.73) |
| Gravidity | | Mean = 1.39, SD = 0.83 | | Mean = 1.45, SD = 0.93 | | <0.001 | 1.07 | (1.05–1.11) |

*(Continued)*

**Table 2.** (Continued)

| Variables | Levels | Term birth | | Preterm birth | | Chi-square test | | |
|---|---|---|---|---|---|---|---|---|
| | | 85457 (93.8%) | | 5645 (6.2%) | | | | |
| | | N | % | N | % | P-Value | Crude OR | 95% CI |
| Gestational weight gain during the first trimester | Recommended | 7934 | 9.28 | 533 | 9.44 | 0.053 | Reference | |
| | >Recommended | 14535 | 17.01 | 1036 | 18.35 | | 1.07 | (0.95–1.18) |
| | <Recommended | 16107 | 18.85 | 1059 | 18.76 | | 0.98 | (0.87–1.09) |
| | Missing | 46881 | 54.86 | 3017 | 53.45 | | 0.96 | (0.87–1.05) |
| Antenatal health care provider | Obstetrician | 73694 | 86.24 | 5104 | 90.42 | <0.001 | Reference | |
| | Midwife | 11763 | 13.76 | 541 | 9.58 | | 0.66 | (0.60–0.72) |
| Alcohol consumption | No | 83881 | 98.16 | 5539 | 98.12 | 0.896 | Reference | |
| | Yes | 1576 | 1.84 | 106 | 1.88 | | 1.02 | (0.83–1.25) |
| Drug (substance) use | No | 83660 | 97.90 | 5470 | 96.90 | <0.001 | Reference | |
| | Yes | 1797 | 2.10 | 175 | 3.10 | | 1.48 | (1.26–1.74) |
| Pre-existing health conditions | None | 70541 | 82.55 | 4259 | 75.45 | <0.001 | Reference | |
| | Yes | 14916 | 17.45 | 1386 | 24.55 | | 1.53 | (1.44–1.63) |
| Pre-existing mental health conditions | No | 73626 | 86.16 | 4720 | 83.61 | <0.001 | Reference | |
| | Yes | 11831 | 13.84 | 925 | 16.39 | | 1.21 | (1.13–1.31) |
| Diabetes during the first trimester | No diabetes | 84938 | 99.39 | 5480 | 97.08 | <0.001 | Reference | |
| | Type I | 226 | 0.26 | 86 | 1.52 | | 5.90 | (4.27–7.53) |
| | Type II | 293 | 0.34 | 79 | 1.40 | | 4.17 | (3.23–5.33) |
| Pregnancy-associated plasma protein A | Normal | 46161 | 54.02 | 3049 | 54.01 | <0.001 | Reference | |
| | Abnormal | 2215 | 2.59 | 324 | 5.74 | | 2.21 | (1.96–2.50) |
| | Missing | 37081 | 43.39 | 2272 | 40.25 | | 0.93 | (0.87–0.98) |
| Nuchal translucency | Normal | 47496 | 55.58 | 3323 | 58.87 | <0.001 | Reference | |
| | Abnormal | 124 | 0.15 | 8 | 0.14 | | 0.92 | (0.41–1.76) |
| | Missing | 37837 | 44.28 | 2314 | 40.99 | | 0.87 | (0.92–0.92) |
| Free beta-subunit of human chorionic gonadotropin | Normal | 3665 | 4.29 | 254 | 4.50 | 0.249 | Reference | |
| | Abnormal | 396 | 0.46 | 34 | 0.60 | | 1.23 | (0.83–1.77) |
| | Missing | 81396 | 95.25 | 5357 | 94.90 | | 0.94 | (0.85–1.08) |

SD: Standard deviation; IVF: *In vitro* fertilization; Pre-existing maternal health conditions shown in S2 Table. Pre-existing mental health conditions shown in S3 Table.

experience PTB. During the second trimester, we examined 35 predictors of PTB. Women who were over 29 years of age, had abnormal concentrations of the assessed proteins, diabetes, hypertensive disorders of pregnancy, women carrying male fetuses, and those with pregnancy complications were more likely than women without these conditions to experience PTB (Table 3).

**Multivariable analysis.** Stepwise logistic regression identified 13 significant predictors during the first trimester (Fig 1). Diabetes (Type I: adjusted odds ratio (AOR): 4.21; 95% confidence interval (CI): 3.23–5.42; Type II: AOR: 2.68; 95% CI: 2.05–3.46) and abnormal pregnancy-associated plasma protein A concentrations (AOR: 2.04; 95% CI: 1.80–2.30) were the most significant predictors of PTB. The following factors were also associated with an increased risk of PTB: pregnancies conceived through IVF, being obese or underweight, maternal drug (substance) use, lower neighbourhood education level, lower neighbourhood immigration level, low maternal height, diabetes, and other pre-existing medical or mental health conditions.

**Table 3. Univariate analyses of associations between each predictor and preterm birth during the second trimester in nulliparous women.**

| Variables | Levels | Term birth 108905 (93.4%) | | Preterm birth 7754 (6.6%) | | Chi-square test | | |
|---|---|---|---|---|---|---|---|---|
| | | N | % | N | % | P-values | OR | 95% CI |
| Age (years) | <20 | 5696 | 5.23 | 322 | 4.15 | <0.001 | 0.81 | (0.72–0.91) |
| | 20–24 | 17681 | 16.24 | 1115 | 14.38 | | 0.90 | (0.84–0.97) |
| | 25–29 | 36048 | 33.10 | 2505 | 32.31 | | Reference | |
| | 30–34 | 34813 | 31.97 | 2598 | 33.51 | | 1.07 | (1.01–1.13) |
| | 35+ | 14667 | 13.47 | 1214 | 15.66 | | 1.19 | (1.10–1.28) |
| Height | <150 cm | 2557 | 2.35 | 232 | 2.99 | <0.001 | 1.28 | (1.10–1.46) |
| | 150 cm—159 cm | 22590 | 20.74 | 1907 | 24.59 | | 1.18 | (1.12–1.26) |
| | 160 cm—169 cm | 60107 | 55.19 | 4270 | 55.07 | | Reference | |
| | ≥170 cm | 23651 | 21.72 | 1345 | 17.35 | | 0.78 | (0.73–0.84) |
| Pre- pregnancy BMI (kg/m$^2$) | Normal | 68198 | 62.62 | 4646 | 59.92 | <0.001 | Reference | |
| | Overweight | 20226 | 18.57 | 1475 | 19.02 | | 1.07 | (1.00–1.14) |
| | Obese | 14648 | 13.45 | 1218 | 15.71 | | 1.22 | (1.14–1.30) |
| | Underweight | 5833 | 5.36 | 415 | 5.35 | | 1.04 | (0.94–1.15) |
| Neighbourhood income quartile | First quartile (lowest) | 30047 | 27.59 | 2182 | 28.14 | 0.350 | 1.01 | (0.92–1.06) |
| | Second quartile | 25068 | 23.02 | 1806 | 23.29 | | Reference | |
| | Third quartile | 26142 | 24.00 | 1866 | 24.06 | | 0.99 | (0.90–1.05) |
| | Fourth quartile (highest) | 27648 | 25.39 | 1900 | 24.50 | | 0.95 | (0.89–1.01) |
| Neighbourhood education quartile | First quartile (lowest) | 27948 | 25.66 | 1878 | 24.22 | 0.020 | 0.94 | (0.88–1.01) |
| | Second quartile | 28630 | 26.29 | 2027 | 26.14 | | Reference | |
| | Third quartile | 27684 | 25.42 | 2012 | 25.95 | | 1.02 | (0.96–1.12) |
| | Fourth quartile (highest) | 24643 | 22.63 | 1837 | 23.69 | | 1.05 | (0.98–1.12) |
| Neighbourhood minority quartile | First quartile (lowest) | 23348 | 21.44 | 1709 | 22.04 | 0.500 | 1.01 | (0.94–1.09) |
| | Second quartile | 18283 | 16.79 | 1317 | 16.98 | | Reference | |
| | Third quartile | 23105 | 21.22 | 1608 | 20.74 | | 0.96 | (0.91–1.04) |
| | Fourth quartile (highest) | 44169 | 40.56 | 3120 | 40.24 | | 0.98 | (0.91–1.04) |
| Neighbourhood immigration quartile | First quartile (lowest) | 24099 | 22.13 | 1822 | 23.50 | 0.040 | 1.09 | (1.01–1.17) |
| | Second quartile | 19780 | 18.16 | 1366 | 17.62 | | Reference | |
| | Third quartile | 24219 | 22.24 | 1683 | 21.70 | | 1.01 | (0.93–1.09) |
| | Fourth quartile (highest) | 40807 | 37.47 | 2883 | 37.18 | | 1.02 | (0.95–1.02) |
| Smoking status | Non-smoker | 98461 | 90.41 | 6906 | 89.06 | <0.001 | Reference | |
| | Smoker | 10444 | 9.59 | 848 | 10.94 | | 1.15 | (1.07–1.24) |
| Ex-smoker | No | 91890 | 84.38 | 6479 | 83.56 | 0.060 | Reference | |
| | Yes | 17015 | 15.62 | 1275 | 16.44 | | 1.06 | (0.99–1.13) |
| Alcohol consumption | No | 106830 | 98.09 | 7590 | 97.88 | 0.210 | Reference | |
| | Yes | 2075 | 1.91 | 164 | 2.12 | | 1.02 | (0.93–1.30) |
| Drug (substance) use | No | 106518 | 97.81 | 7490 | 96.60 | <0.001 | Reference | |
| | Yes | 2387 | 2.19 | 264 | 3.40 | | 1.48 | (1.37–1.78) |
| Number of previous abortions | 0 | 82064 | 75.35 | 5601 | 72.23 | <0.001 | Reference | |
| | 1 | 18748 | 17.22 | 1409 | 18.17 | | 1.10 | (1.03–1.16) |
| | 2 | 5573 | 5.12 | 455 | 5.87 | | 1.19 | (1.08–1.31) |
| | 3+ | 2520 | 2.31 | 289 | 3.73 | | 1.68 | (1.48–1.90) |
| Gravidity | | Mean = 1.42, SD = 0.84 | | Mean = 1.52, SD = 0.96 | | <0.000 | 1.11 | (1.059 1.14) |

*(Continued)*

**Table 3.** (Continued)

| Variables | Levels | Term birth 108905 (93.4%) | | Preterm birth 7754 (6.6%) | | Chi-square test | | |
|---|---|---|---|---|---|---|---|---|
| | | N | % | N | % | P-values | OR | 95% CI |
| Gestational weight gain during the first trimester | Recommended | 9604 | 8.82 | 686 | 8.85 | 0.070 | Reference | |
| | >Recommended | 17942 | 16.47 | 1344 | 17.33 | | 1.05 | (0.95–1.15) |
| | <Recommended | 19556 | 17.96 | 1317 | 16.98 | | 0.94 | (0.85–1.04) |
| | Missing | 61803 | 56.75 | 4407 | 56.84 | | 0.99 | (0.91–1.08) |
| Antenatal health care provider | Obstetrician | 95470 | 87.66 | 7122 | 91.85 | <0.001 | | |
| | Midwife | 13435 | 12.34 | 632 | 8.15 | | 0.63 | (0.58–0.68) |
| Diabetes | No diabetes | 108260 | 99.41 | 7523 | 97.02 | <0.001 | Reference | |
| | Type I | 269 | 0.25 | 123 | 1.59 | | 6.58 | (5.29–8.13) |
| | Type II | 376 | 0.35 | 108 | 1.39 | | 4.13 | (3.31–5.10) |
| Pre-existing health conditions | No | 94116 | 86.42 | 6473 | 83.48 | <0.001 | Reference | |
| | Yes | 14789 | 13.58 | 1281 | 16.52 | | 1.26 | (1.18–1.34) |
| Pre-existing mental health conditions | None | 90395 | 83.00 | 5879 | 75.82 | <0.001 | Reference | |
| | Yes | 18510 | 17.00 | 1875 | 24.18 | | 1.56 | (1.47–1.64) |
| Folic acid use | Yes | 85553 | 78.56 | 6118 | 78.90 | 0.490 | Reference | |
| | No | 23352 | 21.44 | 1636 | 21.10 | | 0.98 | (0.92–1.03) |
| Conception type | Spontaneous | 104362 | 95.83 | 7293 | 94.05 | <0.001 | Reference | |
| | IVF or combination | 2008 | 1.84 | 264 | 3.40 | | 1.88 | (1.64–2.13) |
| | Other | 2535 | 2.33 | 197 | 2.54 | | 1.11 | (0.95–1.28) |
| Pregnancy-associated plasma protein-A | Normal | 58076 | 53.33 | 4122 | 53.16 | <0.001 | Reference | |
| | Abnormal | 2792 | 2.56 | 472 | 6.09 | | 2.38 | (2.14–2.63) |
| | Missing | 48037 | 44.11 | 3160 | 40.75 | | 0.92 | (0.88–0.97) |
| Nuchal translucency | Normal | 59980 | 55.08 | 4539 | 58.54 | <0.001 | Reference | |
| | Abnormal | 158 | 0.15 | 18 | 0.23 | | 1.50 | (0.89–2.38) |
| | Missing | 48767 | 44.78 | 3197 | 41.23 | | 0.86 | (0.82–0.90) |
| Free beta-subunit of human chorionic gonadotropin | Normal | 6195 | 5.69 | 468 | 6.04 | 0.300 | Reference | |
| | Abnormal | 670 | 0.62 | 54 | 0.70 | | 1.07 | (0.78–1.41) |
| | Missing | 102040 | 93.70 | 7232 | 93.27 | | 0.93 | (0.88–1.03) |
| First trimester visit | Yes | 85457 | 78.47 | 5645 | 72.80 | <0.001 | Reference | |
| | No | 10433 | 9.58 | 742 | 9.57 | | 1.07 | (0.99–1.16) |
| | Unknown | 13015 | 11.95 | 1367 | 17.63 | | 1.59 | (1.50–1.69) |
| Intention to breastfeed | Yes | 4514 | 4.14 | 549 | 7.08 | <0.001 | | |
| | No | 104391 | 95.86 | 7205 | 92.92 | | 1.76 | (1.60–1.92) |
| Dimeric inhibin A | Normal | 7415 | 6.81 | 535 | 6.90 | <0.001 | Reference | |
| | Abnormal | 516 | 0.47 | 63 | 0.81 | | 1.69 | (1.27–2.21) |
| | Missing | 100974 | 92.72 | 7156 | 92.29 | | 0.98 | (0.89–1.07) |
| Unconjugated estriol | Normal | 59024 | 54.20 | 4440 | 57.26 | <0.001 | Reference | |
| | Abnormal | 256 | 0.24 | 40 | 0.52 | | 2.07 | (1.46–2.86) |
| | Missing | 49625 | 45.57 | 3274 | 42.22 | | 0.87 | (0.83–0.91) |
| Human chorionic gonadotropin | Normal | 58384 | 53.61 | 4328 | 55.82 | <0.001 | Reference | |
| | Abnormal | 820 | 0.75 | 122 | 1.57 | | 2.01 | (1.64–2.42) |
| | Missing | 49701 | 45.64 | 3304 | 42.61 | | 0.89 | (0.85–0.93) |
| Alpha-fetoprotein | Normal | 58406 | 53.63 | 4190 | 54.04 | <0.001 | Reference | |
| | Abnormal | 1365 | 1.25 | 318 | 4.10 | | 3.42 | (2.85–3.67) |
| | Missing | 49134 | 45.12 | 3246 | 41.86 | | 0.92 | (0.87–0.96) |

(*Continued*)

**Table 3.** (Continued)

| Variables | Levels | Term birth 108905 (93.4%) | | Preterm birth 7754 (6.6%) | | Chi-square test | | |
| | | N | % | N | % | P-values | OR | 95% CI |
|---|---|---|---|---|---|---|---|---|
| Diabetes during the second trimester | No diabetes | 103303 | 94.86 | 6992 | 90.17 | <0.001 | Reference | |
| | Gestational diabetes | 4932 | 4.53 | 524 | 6.76 | | 1.57 | (1.42–1.72) |
| | Type I | 269 | 0.25 | 123 | 1.59 | | 6.75 | (5.43–8.35) |
| | Type II | 376 | 0.35 | 108 | 1.39 | | 4.24 | (3.40–5.24) |
| | Type Unknown | 25 | 0.02 | 7 | 0.09 | | 4.13 | (1.65–9.13) |
| Hypertensive disorder | None | 95411 | 87.61 | 6080 | 78.41 | <0.001 | Reference | |
| | Gestational hypertension | 4812 | 4.42 | 562 | 7.25 | | 1.83 | (1.67–2.01) |
| | Eclampsia | 42 | 0.04 | 24 | 0.31 | | 8.96 | (5.35–14.68) |
| | HELLP | 81 | 0.07 | 112 | 1.44 | | 21.69 | (16.31–28.99) |
| | Preeclampsia | 654 | 0.60 | 288 | 3.71 | | 6.91 | (5.99–7.94) |
| | Unknown | 7905 | 7.26 | 688 | 8.87 | | 1.39 | (1.25–1.48) |
| Infection(s) | No | 79027 | 72.57 | 6055 | 78.09 | <0.001 | Reference | |
| | Yes | 29878 | 27.43 | 1699 | 21.91 | | 1.34 | (1.27–1.42) |
| Medication exposure | No | 20814 | 19.11 | 1444 | 18.62 | <0.001 | Reference | |
| | Vitamins and herbals | 56399 | 51.79 | 3311 | 42.70 | | 0.84 | (0.79–0.90) |
| | Other medication | 31692 | 29.10 | 2999 | 38.68 | | 1.36 | (1.27–1.45) |
| Sex of baby | Female | 53141 | 48.80 | 3365 | 43.40 | <0.001 | Reference | |
| | Male | 55764 | 51.20 | 4389 | 56.60 | | 1.24 | (1.18–1.30) |
| Complications during pregnancy | No complications | 93777 | 86.11 | 2974 | 38.35 | <0.001 | Reference | |
| | Mild-moderate complications | 4538 | 4.17 | 283 | 3.65 | | 1.96 | (1.73–2.22) |
| | Severe complications | 10590 | 9.72 | 4497 | 58.00 | | 13.39 | (12.73–17.08) |

IVF: *In vitro* fertilization; SD: standard deviation; Pre-existing maternal health conditions shown in S2 Table. Pre-existing mental health conditions shown in S3 Table.

During the second trimester, we identified 17 significant predictors related to PTB (Fig 2) using stepwise logistic regression. Many of the selected variables were the same as those selected for the first-trimester model, with slight changes in the odds ratios. Furthermore, severe complications of pregnancy were strongly associated with PTB (AOR: 13.03; 95% CI: 12.21–13.90). Women with abnormal alpha-fetoprotein, those carrying a male fetus, and those who did not attend prenatal classes were at increased odds of PTB. Exposure to medication during pregnancy, including vitamins and herbal supplements, was associated with a decreased risk of PTB.

Machine learning (Boruta) identified 17 and 27 important predictors of PTB during the first and second trimesters, respectively (S5 and S6 Tables). Unlike with logistic regression, machine learning models selected previous abortions (including miscarriages) as the most important predictor of PTB during the first trimester (importance: 28.23 for previous abortions (including miscarriages) vs. 7.79 for diabetes). During the second trimester, complications during pregnancy and hypertensive disorders were the most important predictors of PTB.

**Prediction models and performance measures in the training and validation samples.** In the training sample, we found that random forests had a higher AUC than other models (99%), including logistic regression, which had the third highest AUC (S7 Table). We evaluated the proposed prediction models in the testing sample and found that during the first trimester the AUCs ranged from 53% (random forests) to 60% (artificial neural networks, Fig 3 and Table 4). However, all models had very high negative predictive values of ~95%. During the

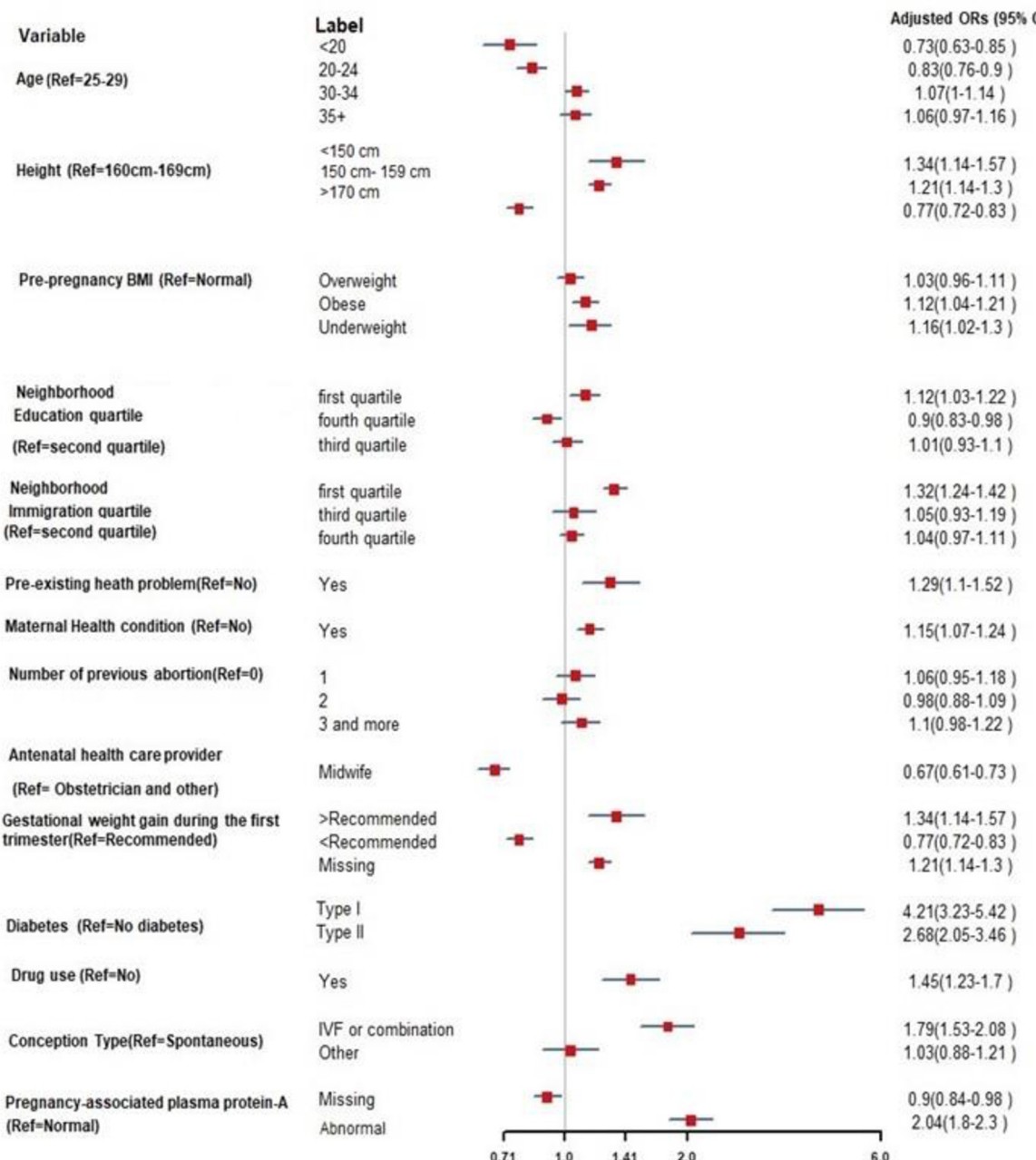

**Fig 1. Selected variables and adjusted odds ratios during the first trimester for prediction of preterm birth in nulliparous women.**
BMI: Body mass index; IVF: *In vitro* fertilization; Ref: Reference group; Pre-existing maternal health conditions shown in S2 Table. Pre-existing mental health conditions shown in S3 Table. Number of previous abortions: includes the number of miscarriages.

second trimester, artificial neural networks had the highest sensitivity of 63% (95% CI: 61–65%, Fig 3 and Table 4), but slightly lower specificity and positive predictive value than logistic regression. Random forests exhibited the lowest sensitivity among the models; however, the positive predictive value of the random forests model was the highest, but still relatively low at 36%.

Overall, there was an increase in the AUC from the first trimester to the second trimester in logistic regression and artificial neural networks (60% vs. 80%). The notable improvement of

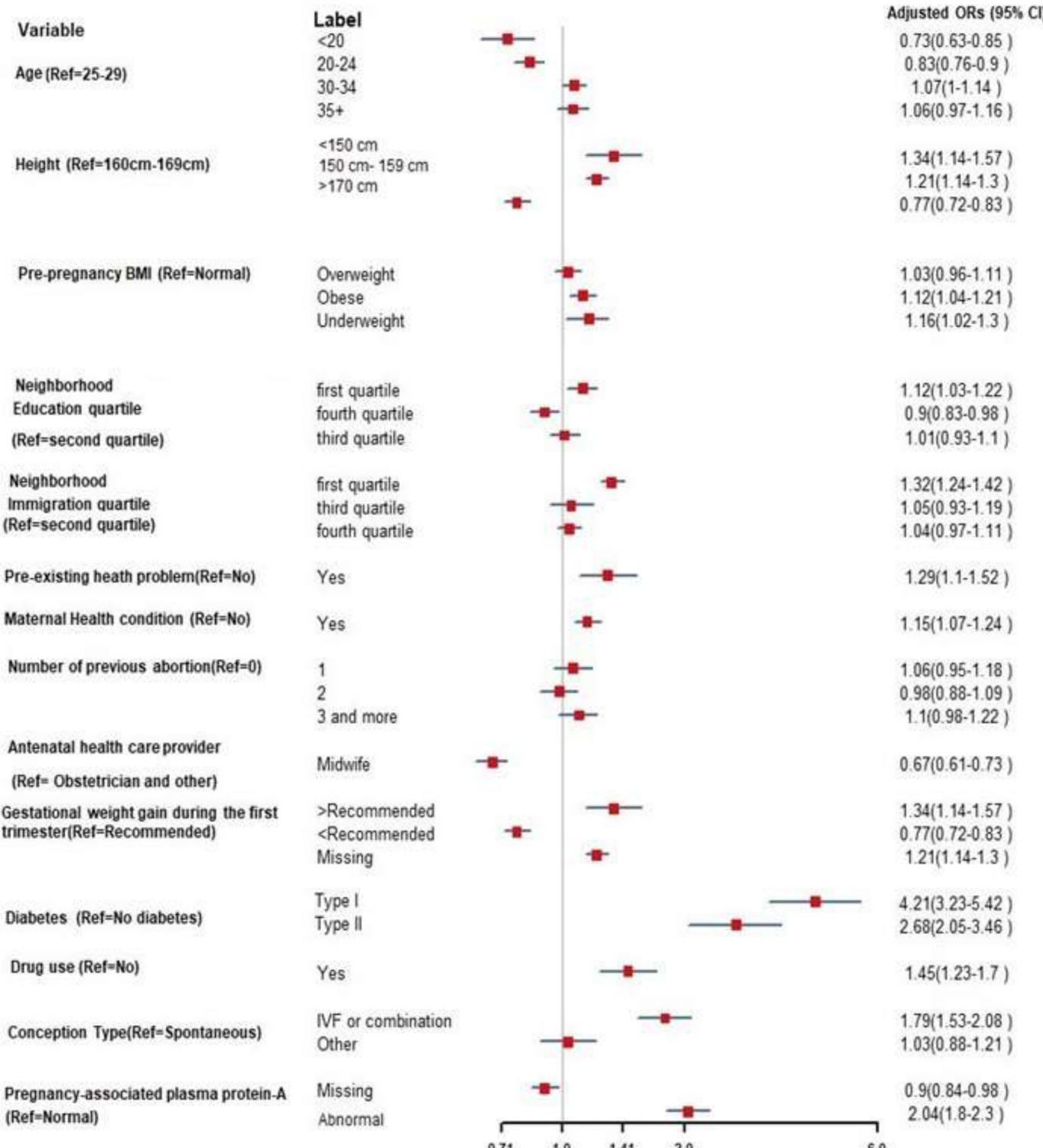

**Fig 2. Selected variables and odds ratios during the second trimester for prediction of preterm birth in nulliparous women.** BMI: Body mass index; IVF: *In vitro* fertilization; Ref: Reference group; Pre-existing maternal health conditions shown in S2 Table. Pre-existing mental health conditions shown in S3 Table. Number of previous abortions: includes the number of miscarriages.

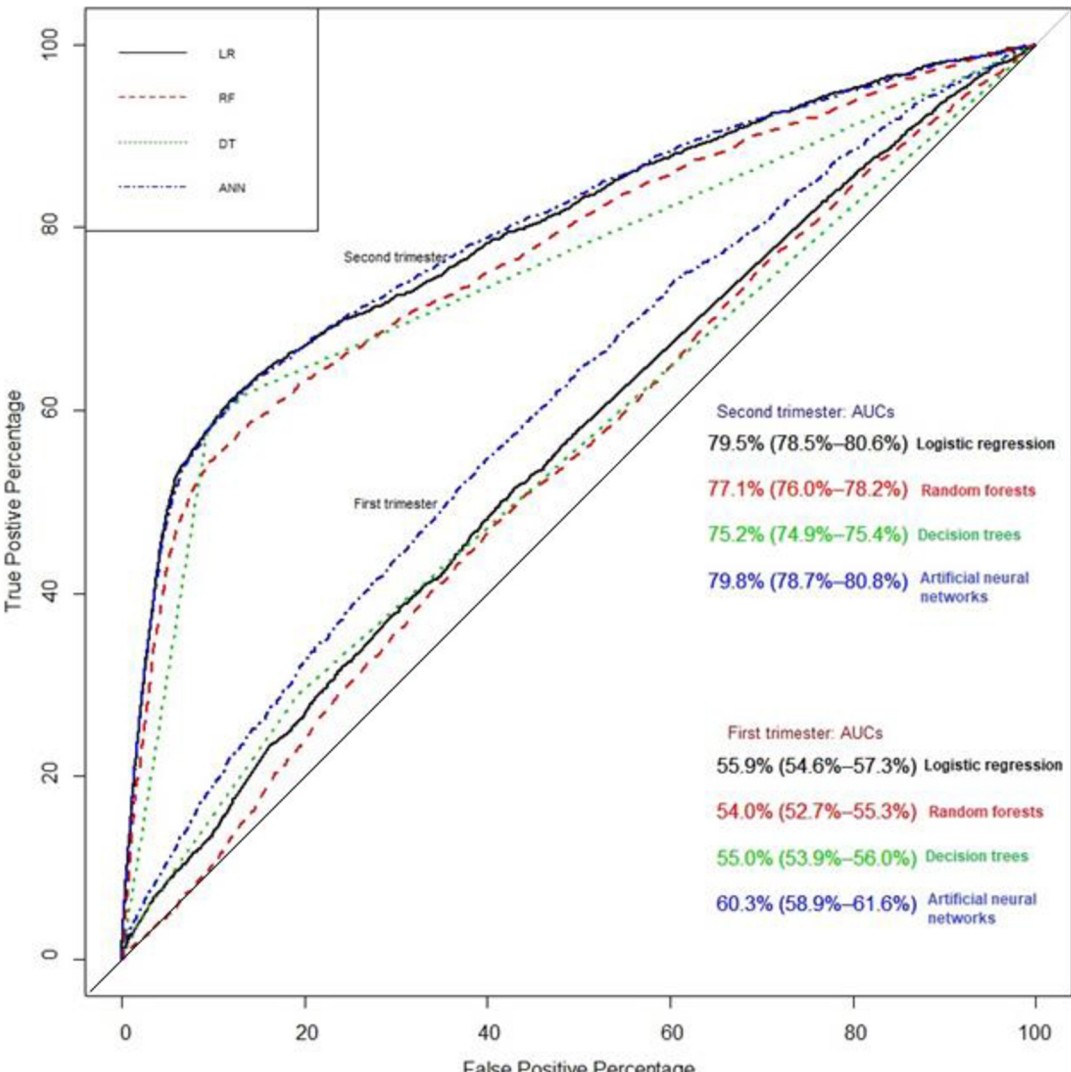

**Fig 3. Comparison of prediction models during the first and second trimester for preterm birth in nulliparous women.**

**Table 4. Predictive power of preterm birth models during the first and second trimesters in nulliparous women.**

| Metric | First trimester | | | | Second trimester | | | |
|---|---|---|---|---|---|---|---|---|
| | Logistic regression | Random forests | Artificial neural networks | Decision trees | Logistic regression | Random forests | Artificial neural networks | Decision trees |
| Sensitivity | 50.2 (47.8–52.4) | 29.4 (26.1–31.6) | 36.0 (34.5–42.3) | 29.2 (27.1–30.8) | 62.2 (60.0–63.4) | 45.2 (44.5–48.5) | 62.7 (61.2–65.4) | 58.1 (55.6–60.2) |
| Specificity | 64.5 (63.1–65.4) | 84.5 (83.0–86.4) | 71.2 (68.2–73.1) | 80.2 (79.5–81.4) | 87.0 (85.5–88.4) | 94.1 (93.8–95.2) | 84.6 (83.1–86.5) | 90.1 (89.2–91.4) |
| Positive predictive value | 8.5 (8.1–9.3) | 11.4 (9.1–12.2) | 11.3 (8.3–13.4) | 9.2 (8.5–10.4) | 25.2 (24.5–26.3) | 36.0 (35.3–38.4) | 23.2 (21.3–23).3 | 29.1 (27.1–29.2) |
| Negative predictive value | 95.5 (94.4–95.3) | 95.2 (94.9–96.1) | 95.0 (94.1–95.3) | 94.2 (93.9–95.2) | 97.3 (96.3–98.3) | 96.2 (95.6–97.2) | 97.0 (96.5–98.2) | 97.2 (96.1–98.4) |

All values of percentages; 95% confidence intervals are given in parentheses.

the AUC to 80% with artificial neural networks and logistic regression was due to the addition of complications during pregnancy (S1 and S3 Figs). All models provided negative predictive value of ~97% during the second trimester. In a sensitivity analysis, we compared the predictive power of all models without complications during pregnancy, and found that the AUC ranged from 58% (decision trees) to 65% (artificial neural networks, S1 Fig).

### Prediction of spontaneous PTB

For models predicting spontaneous PTB, during the first trimester the AUC ranged from 55% (random forests) to 59% (logistic regression, S2 Fig). During the second trimester, AUC ranged from 58% (decision trees) to 64% (logistic regression, S3 Fig). Both machine learning and logistic regression generated negative predictive values of approximately 94% for spontaneous PTB during the first and second trimesters (S8 Table). We emphasize that pregnancy complications, hypertensive disorder, and other medically induced PTB were not included in these analyses.

### Discussion

We used population-based data to predict PTB in nulliparous women using logistic regression and machine learning approaches during the first and second trimesters. We found that diabetes mellitus, a history of spontaneous or therapeutic abortions, and abnormal pregnancy-associated plasma protein A concentrations were the strongest predictors for PTB during the first trimester. Thirteen selected predictors yielded a maximum AUC of 60% with artificial neural networks, thus providing poor prediction of PTB during the first trimester, even using machine learning approaches. During the second trimester, 17 variables were significantly associated with PTB, among which complications during pregnancy had the highest AOR (13.03; 95% CI: 12.21–13.9). During the second trimester, the AUC increased from 65% (95% CI: 63–66%) to 80% (95% CI: 79–81%) with the inclusion of complications during pregnancy, which is a moderate predictor [50] of PTB.

Machine learning identified more variables associated with PTB than logistic regression in our data set. During the first trimester, machine learning identified previous abortions (which includes miscarriages) as the strongest predictor of PTB, while logistic regression identified diabetes as the strongest predictor. A history of prior abortions (including miscarriages) may be a more important predictor of PTB because the incidence of prior abortions was substantially higher than that of diabetes.

We found that conventional logistic regression and machine learning had comparable performance for prediction of PTB. Other studies comparing machine learning methods to conventional logistic regression for the prediction of a variety of clinical conditions showed that in general, no single method consistently provided the best prediction [51–58]. Although logistic regression is a frequently used method, it requires linearity and independence between the predictors. Conversely, machine learning is a non-parametric approach that can handle complex and non-linear models.

There was a significant decrease in the AUC between the training and the testing data, possibly due to the overfitting problem of machine learning methods [54]. Specifically, random forests are "greedy", and thus, try to minimize the error in the training sample, which may cause overfitting (high performance in training but lower performance in the validation sample, as we observed in our models) [30].

Accurate prediction of PTB in nulliparous women has been lacking. Woolery and Grzymala [55] found machine learning had 53–88% accuracy in predicting PTB. Using data mining methods, Goodwin *et al.* found that seven demographic variables produced an AUC of 72%

[10]. In contrast, Grobman *et al*. [12] found that logistic regression provided poor performance (AUC, 63%) for prediction of PTB in nulliparous women with a short cervix. Catley *et al*. [15] explored artificial neural networks for the prediction of PTB in high-risk pregnant women and found model sensitivity of 20% before 22 weeks of gestation. Weber *et al*. [13] recently applied machine learning to predict early (<32 weeks) spontaneous PTB among nulliparous women and found an AUC of only 63–65%, similar to Courtney *et al*. [56] (AUC, 60%) using logistic regression and a support vector machine approach.

## Strengths of the study

Our study had several strengths. Firstly, our models generated high negative predictive values, higher than fetal fibronectin for spontaneous PTB [57], and thus may lead to reduction in unnecessary resource use [58]. Secondly, we considered a wide range of variables available in standard clinical care databases (e.g., proteins for screening for Down syndrome or placental diseases, gestational weight gain) that were not considered in previous studies. Another strength of the current work is the consideration of different time points (first and second trimesters) for the prediction of PTB. In addition, we evaluated a relatively large cohort, particularly compared to many of the previous studies [8–14]. We considered multiple methods for variable selection and prediction to maximize accuracy. We addressed several limitations of previous studies in this area: Courtney *et al*. [56] found that logistic regression and machine learning models based on demographic data were not able to predict PTB adequately (AUC, 60%). Those authors suggested that prenatal demographic factors such as maternal health behaviors and medical history could be used to construct accurate models, and thus, we included such factors in our study. By performing a large cohort study, we also addressed the "lack of data" problem identified in the work of Lee *et al*. [11]. We applied multiple imputation (repeated ten times), which is a robust technique for handling missing data [48]. Unlike Fergue *et al*. [16], we used random oversampling in the training set only, thus the AUC from our models was generated from clinical data and not artificial samples.

## Limitations

Our study also has several limitations, including the low predictive power of the proposed models, particularly during the first trimester. The predictive ability of all models strongly depends on the predictor variables [30]. Although we had a large number of variables and a relatively large number of subjects, one of the limitations of our prediction models was the lack of information on the interventions used for pregnancies at high risk of PTB. However, data suggest relatively low rates of use of such preventive measures in our study population [59]. We categorized PTB as <37 or ≥37 weeks of gestation, which may lead to loss of statistical power [60]. Further, binary categorization collapses all types of PTB in one group despite different rates of neonatal mortality and morbidity for each category of PTB [61] and despite potentially different predictors of extremely PTB compared to PTB overall. Although low pregnancy-associated plasma protein A concentraion is associated with trisomies which themselves are associated with preterm birth, the majority of such cases are in euploid pregnancies [62–66]. Finally, we were unable to examine ultrasonographic measurement of the uterine cervix, which is a strong predictor of PTB [67] as it is not available in the BORN database.

## Conclusion

Including data from the second trimester improved prediction power to a moderate level of 80% AUC by both logistic regression and machine learning. However, developing an accurate

prediction model during the first trimester will require further investigation. Inclusion of data from additional biomarkers may increase prediction accuracy.

## Supporting information

**S1 Fig. Receiver operating characteristic curves for second-trimester prediction models without the "complications during pregnancy" variable in the validation sample.**
(DOCX)

**S2 Fig. Receiver operating characteristic curves for first-trimester prediction models for spontaneous preterm birth in the validation sample.**
(DOCX)

**S3 Fig. Receiver operating characteristic curves for second-trimester prediction models for spontaneous preterm birth in the validation sample.**
(DOCX)

**S1 Table. Definitions of neighbourhood income, immigration, education, and minority quartiles.**
(DOCX)

**S2 Table. Pre-existing maternal health conditions.**
(DOCX)

**S3 Table. Pre-existing mental health conditions.**
(DOCX)

**S4 Table. Cut-off points for nuchal translucency and protein concentrations.**
(DOCX)

**S5 Table. Variables selected by the machine learning algorithm for prediction of preterm birth during the first trimester in nulliparous women.**
(DOCX)

**S6 Table. Variables selected by the machine learning algorithm for prediction of preterm birth during the second trimester in nulliparous women.**
(DOCX)

**S7 Table. Optimal hyperparameters, sensitivity, specificity, and area under the receiver operating characteristic curve in training samples.**
(DOCX)

**S8 Table. Predictive power of spontaneous preterm birth models during the first and second trimesters in the testing data.**
(DOCX)

## Acknowledgments

We greatly appreciate the assistance of our Associate Editor and two anonymous referees for careful reading and valuable suggestions on our manuscript that significantly improved the presentation of the paper.

## Author Contributions

**Formal analysis:** Reza Arabi Belaghi.

**Methodology:** Reza Arabi Belaghi.

**Software:** Reza Arabi Belaghi.

**Supervision:** Joseph Beyene, Sarah D. McDonald.

**Writing – original draft:** Reza Arabi Belaghi.

**Writing – review & editing:** Joseph Beyene, Sarah D. McDonald.

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
