## [Decision Letter · Decision Letter 0]

18 Jan 2021

PONE-D-20-30837

Prediction of Preterm Birth in Nulliparous Women Using Logistic Regression and Machine Learning

PLOS ONE

Dear Dr. McDonald,

Thank you for submitting your manuscript to PLOS ONE. After careful consideration, we feel that it has merit but does not fully meet PLOS ONE’s publication criteria as it currently stands. Therefore, we invite you to submit a revised version of the manuscript that addresses the points raised during the review process.

In addition to the issues raised by the reviewers, please provide the code for AI if possible.

It is an interesting approach and might be further improved with new data.

We look forward to receiving your revised manuscript.

Kind regards,

Pal Bela Szecsi, M.D. D.M.Sci.

Academic Editor

PLOS ONE

Journal Requirements:

3. Please ensure that you refer to Figure 3 in your text as, if accepted, production will need this reference to link the reader to the figure.

Reviewers' comments:

Reviewer's Responses to Questions

**Comments to the Author**

1. Is the manuscript technically sound, and do the data support the conclusions?

Reviewer #1: Yes

Reviewer #2: Yes

2. Has the statistical analysis been performed appropriately and rigorously? 

Reviewer #1: Yes

Reviewer #2: Yes

3. Have the authors made all data underlying the findings in their manuscript fully available?

Reviewer #1: Yes

Reviewer #2: Yes

4. Is the manuscript presented in an intelligible fashion and written in standard English?

Reviewer #1: No

Reviewer #2: Yes

5. Review Comments to the Author

Reviewer #1: In this study Authors constructed by using logistic regression analysis and machine learning technique an algorithm to predict preterm labor defined as < 37 weeks. The argument is of interest, the number of women considered relevant and an elegant statistical approach was used. So I would like to congratulate with Authors for their effort

My comments are as follows

1) did Authors differentiate spontaneous from iatrogenic preterm delivery? This is of crucial since women with pregestational diseases (diabetes) or developing medical complications are frequently induced preterm and this may flaw the algorithm

2)although stated as a limitation I suggest Authors to perform their analysis also at earlier gestational age (e.g. < 34 and or 32 weeks) that are more clinical significant

3)it should be acknowledged that data on ultrasonographic measurement of the uterine cervix are missing since at present it is considered the powerful predictive variables.

Reviewer #2: In this manuscript, Belaghi et al use a database of nulliparous women who delivered in Ontario, Canada to predict PTB using both logistic regression and machine learning techniques. They found that using data available from the second trimester improved their prediction models using both approaches. The paper is well-written and easy to understand. However, several important questions arise from this study in its current form:

1. Spontaneous PTB: How was this defined? This is not clear from the manuscript. This should be clarified. Further, as the authors allude but do not directly discuss, PTB can be broadly classified into provider-initiated PTB and spontaneous PTB. The pathophysiology of spontaneous PTB is very different than that of provider-initiated PTB. Although this study is by no means the first study to group PTB broadly into one category, it should directly address the reality that PTB has many phenotypes and that a prediction algorithm that is trying to predict all PTB inherently has many limitations. An algorithm that predicts spontaneous PTB may be of greater utility and greater accuracy than an algorithm that tries to predict both spontaneous PTB and HELLP syndrome necessitating provider-initiated delivery. Further, in the abstract, the authors compare their model to the negative predictive value of a fetal fibronectin test. A fetal fibronectin test is ONLY used to predict spontaneous PTB, not all PTB. Consequently, this comparison is of little utility.

Further, what percentage of PTB included in this study was spontaneous? This is not clear from the manuscript. If possible, the authors should provider information on the various phenotypes of PTB and how they were ascertained. This information is of significant clinical utility.

2. PAPP-A: In the abstract, the authors mention "abnormal pregnancy-associated plasma protein-A contractions" as being strongly associated with PTB. However, how a PAPP-A contraction was defined is unclear, as this contraction is never mentioned again. Is this meant to read concentration, not contraction?

3. Complications during pregnancy: No definition of moderate complications is provided in the manuscript. Additionally, what percentage of women had each of the severe complications listed on page 6 is not clear. The only clarity regarding this variable is provided on page 6: "The variable, complications during pregnancy, had more than 600 categories and we classified those data into three groups based on the expert opinion of our in-home maternal-fetal specialist, including no complications, moderate complications, and severe complications (including hypertensive disorder, placental abnormalities, and maternal complications during this pregnancy, such as antepartum bleeding)." As severe complications of pregnancy were highly associated with PTB, it would be helpful to better understand this variable. Further, if possible, these complications should be separated and included in the model, as one would expect preeclampsia and HELLP are more likely to lead to provider-initiated PTB and antepartum bleeding to be associated with abruption and preterm labor, which would likely lead to spontaneous PTB.

4. Aneuploidy: This study does not directly address aneuploidy or trisomy pregnancies. However, the most significant predictors of PTB in this study were diabetes and PAPP-A. Diabetes and low PAPP-A are both associated with trisomy pregnancies, and trisomy pregnancies have increased risks of PTB. Consequently, this should be addressed/clarified in this manuscript.

5. Grammar: Please carefully review the manuscript at length for typos. Below are several that were identified on my review:

- A parenthesis is missing after "(Supplemental Table 2" on page 6.

- In the last paragraph on page 9, the first sentence should include an "s" after "other model."

- An extra parenthesis should be removed after "(logistic regression, Supplementary Figure 2))" and "(logistic regression, Supplementary Figure 3))" on page 10.

- "Table S8" should be renamed "Supplemental Table 8" to be consistent with the rest of the manuscript.

6. PLOS authors have the option to publish the peer review history of their article (what does this mean?). If published, this will include your full peer review and any attached files.

Reviewer #1: **Yes: **Giuseppe Rizzo

Reviewer #2: **Yes: **Katelyn J Rittenhouse, MD

---

## [Author Response · Author response to Decision Letter 0]

8 Mar 2021

Dear Editor and Reviewers,

We greatly appreciate your careful reading of our manuscript and the helpful feedback you have provided. We have revised the manuscript based on your comments, as detailed below. We thank you again for your valuable time and we hope that you find our revised manuscript acceptable for publication. Our responses to your comments are given in Blue font. We also added the requested revisions in the text in the Blue font. We also reformatted the paper according to the journal guidelines. 

Reviewer #1: In this study Authors constructed by using logistic regression analysis and machine learning technique an algorithm to predict preterm labor defined as < 37 weeks. The argument is of interest, the number of women considered relevant and an elegant statistical approach was used. So I would like to congratulate with Authors for their effort

My comments are as follows

1) did Authors differentiate spontaneous from iatrogenic preterm delivery? This is of crucial since women with pregestational diseases (diabetes) or developing medical complications are frequently induced preterm and this may flaw the algorithm

We thank the reviewer for this important comment. We did examine spontaneous PTB as a secondary outcome, excluding medically induced PTB and women with PPROM. Results from this analysis are presented in the last paragraph of the Results section and are included below for your convenience. 

Prediction of spontaneous PTB

For models predicting spontaneous PTB, during the first trimester the AUC ranged from 55% (random forests) to 59% (logistic regression, Supplementary Figure 2). During the second trimester, AUC ranged from 58% (decision trees) to 64% (logistic regression, Supplementary Figure 3). Both machine learning and logistic regression generated negative predictive values of approximately 94% for spontaneous PTB during the first and second trimesters (Supplementary Table 8). We emphasize that pregnancy complications, hypertensive disorder, and other medically induced PTB were not included in these analyses.

2) Although stated as a limitation I suggest Authors to perform their analysis also at earlier gestational age (e.g. < 34 and or 32 weeks) that are more clinical significant

We previously developed separate prediction models for preterm birth <32 weeks and <28 weeks. Because of the differing prevalence, and risk factors, between these outcomes and PTB <37 weeks and because the analyses for the present manuscript are focused on nulliparous women, we chose to publish the models for earlier PTB outcomes in a separate manuscript, which is currently in press elsewhere. 

3) It should be acknowledged that data on ultrasonographic measurement of the uterine cervix are missing since at present it is considered the powerful predictive variables.

We thank the reviewer for this suggestion. Although ultrasonographic measurement of the uterine cervix during the second trimester is indeed a strong predictor of preterm birth, the BORN database does not include data on such measurements. We now discuss this in the limitations section (last paragraph of the Discussion). 

Reviewer #2: In this manuscript, Belaghi et al use a database of nulliparous women who delivered in Ontario, Canada to predict PTB using both logistic regression and machine learning techniques. They found that using data available from the second trimester improved their prediction models using both approaches. The paper is well-written and easy to understand. However, several important questions arise from this study in its current form:

1. Spontaneous PTB: How was this defined? This is not clear from the manuscript. This should be clarified. 

We defined spontaneous PTB using the definition from Maghsouldu et al. (2019), as follows: not “induced”, not “caesarean section” and not “augmented labor We now clarify the definition of spontaneous PTB in the outcome subsection (second paragraph of the Methods).

Maghsoudlou, S., Yu, Z. M., Beyene, J., & McDonald, S. D. (2019). Phenotypic classification of preterm birth among nulliparous women: a population-based cohort study. Journal of Obstetrics and Gynaecology Canada, 41(10), 1423-1432.

Further, as the authors allude but do not directly discuss, PTB can be broadly classified into provider-initiated PTB and spontaneous PTB. The pathophysiology of spontaneous PTB is very different than that of provider-initiated PTB. Although this study is by no means the first study to group PTB broadly into one category, it should directly address the reality that PTB has many phenotypes and that a prediction algorithm that is trying to predict all PTB inherently has many limitations. An algorithm that predicts spontaneous PTB may be of greater utility and greater accuracy than an algorithm that tries to predict both spontaneous PTB and HELLP syndrome necessitating provider-initiated delivery. 

In addition to our principal analyses, we also developed prediction models for spontaneous PTB as a secondary outcome. The results from this analysis are reported in the last paragraph of the Results section. 

Further, in the abstract, the authors compare their model to the negative predictive value of a fetal fibronectin test. A fetal fibronectin test is ONLY used to predict spontaneous PTB, not all PTB. Consequently, this comparison is of little utility.

In line with the reviewer’s suggestion, we have removed the comparison between our predictive model for overall PTB and FFN from the abstract, and we now refer to FFN only in connection with spontaneous PTB.

Further, what percentage of PTB included in this study was spontaneous? This is not clear from the manuscript. If possible, the authors should provider information on the various phenotypes of PTB and how they were ascertained. This information is of significant clinical utility. 

There were a total of 3468 spontaneous preterm births in our analytic data set, accounting for 46.7% of all PTB (3468/7430) and yielding a spontaneous PTB rate of 6.9% (3468/(46213+3468)). We now clarify this in the footnote to Table 1. We ascertained spontaneous PTB using the definition from Maghsouldu et al., as discussed in our response to point 1 above and in the second paragraph of the Methods section.

Maghsoudlou, S., Yu, Z. M., Beyene, J., & McDonald, S. D. (2019). Phenotypic classification of preterm birth among nulliparous women: a population-based cohort study. Journal of Obstetrics and Gynaecology Canada, 41(10), 1423-1432.

2. PAPP-A: In the abstract, the authors mention "abnormal pregnancy-associated plasma protein-A contractions" as being strongly associated with PTB. However, how a PAPP-A contraction was defined is unclear, as this contraction is never mentioned again. Is this meant to read concentration, not contraction?

We thank the reviewer for drawing our attention to this error. This word was indeed intended to be “concentration” and we have changed it accordingly. 

3. Complications during pregnancy: No definition of moderate complications is provided in the manuscript. Additionally, what percentage of women had each of the severe complications listed on page 6 is not clear. The only clarity regarding this variable is provided on page 6: "The variable, complications during pregnancy, had more than 600 categories and we classified those data into three groups based on the expert opinion of our in-home maternal-fetal specialist, including no complications, moderate complications, and severe complications (including hypertensive disorder, placental abnormalities, and maternal complications during this pregnancy, such as antepartum bleeding)." As severe complications of pregnancy were highly associated with PTB, it would be helpful to better understand this variable. Further, if possible, these complications should be separated and included in the model, as one would expect preeclampsia and HELLP are more likely to lead to provider-initiated PTB and antepartum bleeding to be associated with abruption and preterm labor, which would likely lead to spontaneous PTB.

We thank the reviewer for bringing this point to our attention. Categorization of complications as mild-moderate complications versus severe was based on expert maternal-fetal input (Dr. Sarah McDonald). We now clarify this in the last paragraph of the Predictors subsection, just above Statistical Analysis.

The distribution of complications during pregnancy, which we have added to the end of Table 1, is as follows: 

Complications during pregnancy N Percent

No complications 90302 79.94

Mild-Moderate complications 14255 12.62

Severe complications 4676 4.14

Missing 3730 3.30

This variable was not included in the first-trimester prediction models for overall or spontaneous PTB, whereas it was included in the second-trimester prediction model for overall PTB but not for the spontaneous PTB. We report on the significant predictor variables included in the different models beginning in paragraph 4 of the Results.

We examined the predictive power of all models without complications during pregnancy as a sensitivity analysis, reported in the second to last paragraph of the Results. The AUC in models without complications during pregnancy ranged from 58% (decision trees) to 65% (artificial neural networks, Supplementary Figure 1).

4. Aneuploidy: This study does not directly address aneuploidy or trisomy pregnancies. However, the most significant predictors of PTB in this study were diabetes and PAPP-A. Diabetes and low PAPP-A are both associated with trisomy pregnancies, and trisomy pregnancies have increased risks of PTB. Consequently, this should be addressed/clarified in this manuscript.

We have added this to the limitations section of the discussion section in line with the reviewer’s suggestion. 

5. Grammar: Please carefully review the manuscript at length for typos. Below are several that were identified on my review:

- A parenthesis is missing after "(Supplemental Table 2" on page 6.

- In the last paragraph on page 9, the first sentence should include an "s" after "other model."

- An extra parenthesis should be removed after "(logistic regression, Supplementary Figure 2))" and "(logistic regression, Supplementary Figure 3))" on page 10.

- "Table S8" should be renamed "Supplemental Table 8" to be consistent with the rest of the manuscript.

We thank the reviewer for drawing our attention to these errors. Our manuscript has now undergone additional editorial review, through which we have addressed these and other points.

---

## [Decision Letter · Decision Letter 1]

10 May 2021

Prediction of Preterm Birth in Nulliparous Women Using Logistic Regression and Machine Learning

PONE-D-20-30837R1

Dear Dr. McDonald,

We’re pleased to inform you that your manuscript has been judged scientifically suitable for publication and will be formally accepted for publication once it meets all outstanding technical requirements.

Kind regards,

Pal Bela Szecsi, M.D. D.M.Sci.

Academic Editor

PLOS ONE

Additional Editor Comments (optional):

Please in the poof correct some misspellings (ie. diabetes i fig 1)

**Comments to the Author**

1. If the authors have adequately addressed your comments raised in a previous round of review and you feel that this manuscript is now acceptable for publication, you may indicate that here to bypass the “Comments to the Author” section, enter your conflict of interest statement in the “Confidential to Editor” section, and submit your "Accept" recommendation.

Reviewer #1: All comments have been addressed

2. Is the manuscript technically sound, and do the data support the conclusions?

Reviewer #1: Yes

3. Has the statistical analysis been performed appropriately and rigorously? 

Reviewer #1: Yes

4. Have the authors made all data underlying the findings in their manuscript fully available?

Reviewer #1: Yes

5. Is the manuscript presented in an intelligible fashion and written in standard English?

Reviewer #1: Yes

6. Review Comments to the Author

Reviewer #1: (No Response)

7. PLOS authors have the option to publish the peer review history of their article (what does this mean?). If published, this will include your full peer review and any attached files.

Reviewer #1: **Yes: **Giuseppe Rizzo

---

## [Editor Report · Acceptance letter]

18 Jun 2021

PONE-D-20-30837R1 

Prediction of Preterm Birth in Nulliparous Women Using Logistic Regression and Machine Learning 

Dear Dr. McDonald:

I'm pleased to inform you that your manuscript has been deemed suitable for publication in PLOS ONE. Congratulations! Your manuscript is now with our production department. 

Kind regards, 

on behalf of

Dr. Pal Bela Szecsi 

Academic Editor

PLOS ONE